# Mixed Segmental Uniparental Disomy of Chromosome 15q11-q1 Coexists with Homozygous Variant in *GNB5* Gene in Child with Prader–Willi and Lodder–Merla Syndrome

**DOI:** 10.3390/genes16060689

**Published:** 2025-06-05

**Authors:** Tomasz Marczyk, Maria Libura, Beata Wikiera, Magdalena Góralska, Agnieszka Pollak, Marlena Telenga, Rafał Płoski, Robert Śmigiel

**Affiliations:** 1Department of Pediatric Neurology, T. Marciniak Lower Silesian Specialist Hospital-Emergency Medicine Center, 54-049 Wrocław, Poland; 2Department of Public Health and Social Medicine, Medical University of Gdansk, 80-211 Gdansk, Poland; maria.libura@gumed.edu.pl; 3Department of Pediatrics, Endocrinology, Diabetology and Metabolic Diseases, Medical University of Wroclaw, 50-368 Wrocław, Poland; beata.wikiera@umw.edu.pl (B.W.); marlena.telenga@student.umw.edu.pl (M.T.); 4Department of Endocrinology, Medical University of Warsaw, 02-091 Warsaw, Poland; magdalena.goralska@wum.edu.pl; 5Department of Medical Genetics, Medical University of Warsaw, 02-106 Warsaw, Poland; agnieszka.pollak@wum.edu.pl (A.P.); rploski@wp.pl (R.P.)

**Keywords:** Prader–Willi syndrome, Lodder–Merla syndrome, maternal uniparental disomy, *GNB5*, IDDCA, LADCI

## Abstract

Background: Uniparental disomy (UPD) refers to the condition in which both chromosomes (or part of chromosome) of a pair are inherited from the same parent. There are two types of UPD: uniparental isodisomy (both chromosomes inherited from one parent are identical copies) and uniparental heterodisomy (two different chromosomes are inherited from one parent). UPD presents two primary developmental risks: recessive trait inheritance or an imprinting disorder. These risks may coexist, leading to an ultra-rare comorbidity. Managing the comorbidities associated with rare diseases presents unique clinical challenges. Results: The existence of such phenomena is evidenced by our case report of a boy who was ultimately diagnosed with two rare diseases: Prader–Willi syndrome (PWS), due to the maternal uniparental disomy of chromosome 15 (UPD), and autosomal recessive Lodder–Merla type 1 syndrome, linked to a novel pathogenic variant in the G protein subunit β *5* (*GNB5*) gene, as detailed in this paper. Conclusions: An unusual or severe phenotype in a patient diagnosed with PWS should invariably prompt the consideration of a comorbid genetic disease attributable to genes located in the PWS critical region of chromosome 15q, or elsewhere on chromosome 15. In cases of epileptic encephalopathy with cardiac arrhythmia, prompt consultation with a cardiologist and comprehensive genetic testing are essential to reduce the risks associated with untreated arrhythmia and ensure the provision of appropriate and safe anti-epileptic therapy. The presented case provides further support for the hypothesis that uniparental disomy may serve as an underlying cause of Lodder–Merla syndrome. This underscores the significance of comprehensive genetic testing, encompassing parental testing and familial cascade testing (in selected cases where there is consanguinity, or the likelihood of close common ancestral background between partners) to establish the recurrence risk.

## 1. Introduction

Prader–Willi syndrome (PWS; OMIM: 176270) is a well-known neurodevelopmental syndrome attributed to the loss of function of paternal copies of maternally imprinted genes located in the q11-q13 region of chromosome 15 [1,2]. Patients with PWS present with distinctive dysmorphic features and multiple developmental, behavioral, intellectual, cognitive and endocrine issues [2]. In this case report, we present the case of a boy who was initially diagnosed with Prader–Willi syndrome due to the maternal uniparental disomy of chromosome 15 [3,4,5]. The boy exhibited an unusual neurological and behavioral phenotype: “myopathic” facies, bilateral ptosis, open mouth, weak cry, poor facial expression due to facial muscle weakness, respiratory failure, short proximal limb segments, no visual contact, no social smile, slow pupil reaction to light, bilateral horizontal nystagmus, an absence of visual fixation, profound global hypotonia, poor fine and gross motor skills and seizures. The severe clinical presentation could not be fully explained by the Prader–Willi syndrome diagnosis alone [5,6]. Consequently, genetic testing was extended to whole-exome sequencing (WES). Whole-exome sequencing identified an additional autosomal recessive condition associated with a novel pathogenic variant in the G protein subunit β 5 (*GNB5*) gene. Ultimately, the co-occurrence of Prader–Willi syndrome and autosomal recessive Lodder–Merla syndrome attributed to the *GNB5* gene was confirmed. Lodder–Merla syndrome (OMIM: 617173; OMIM: 617182) is characterized by delayed psychomotor development, profound intellectual disability, absent speech, bradycardia and other cardiac arrhythmias (e.g., atrioventricular block, atrial tachycardia, atrial fibrillation and sinus tachycardia), visual abnormalities (strabismus, myopic refractive errors and horizontal or vertical nystagmus), epilepsy, hypotonia, and gastric reflux. According to the available literature, the co-occurrence of Prader–Willi and Lodder–Merla syndromes has not previously been reported. The case of this patient was presented during a session of the Joint Congress of the European Society for Paediatric Endocrinology (ESPE) and the European Society of Endocrinology (ESE) 2025 as a poster [7].

## 2. Case Report

The male infant, aged 14 months, was born to unrelated, healthy parents via cesarean section at the 38th week of gestation due to breech presentation. The mother observed reduced fetal movement. The baby was subsequently diagnosed with intrauterine growth restriction based on ultrasound-estimated fetal measurements. No additional complications were reported during the course of pregnancy. The infant weighed 2110 g at birth (below 3rd percentile according to WHO growth standards [8]), with a length of 48 cm (15th percentile according to [8]), head circumference of 34 cm (15–50 percentile according to [8], and an Apgar score of 7 at 1 min and 10 at 10 min. A physical examination conducted post-birth revealed bilateral cryptorchidism, with no other structural defects. However, marked hypotonia, a weak sucking reflex, and recurrent severe apneas became apparent shortly after birth. In the first week of life the patient required respiratory assistance, nasal CPAP and, subsequently, mechanical ventilation. On the ninth day of life he was transferred to the newborn intensive care unit (NICU) of the hospital due to his severe general condition, bilateral pneumonia and respiratory failure. While respiratory assistance was no longer necessary, the patient continued to exhibit persistent hypotonia, a weak cry, decreased limb movement, and a weak sucking reflex. Nasogastric feeding was initiated. At four months of age, the patient began to exhibit infantile seizures. The patient experienced seizures on a daily basis, with a frequency of four to six episodes per day, each lasting approximately three minutes. Subsequently, the boy presented with “myopathic” facies (bilateral ptosis, open mouth, and poor facial expression due to facial muscle weakness), short proximal limb segments, and an anterior fontanelle measuring 2.5 × 3 cm at four months of age. Additionally, he manifested no visual contact, no social smile, a slow pupil reaction to light, and bilateral horizontal nystagmus. The patient demonstrated an absence of fixation, profound global hypotonia, and poor fine and gross motor skills, with the exception of hand-to-mouth movement. There was no head control in both prone and supine positions. Tendon reflexes in the upper limbs were weak, while those in the lower limbs were absent. The patient exhibited no grasp reflexes and mild, removable contractures of the fingers. The initial MRI of the brain appeared normal, yet an EEG registered an atypical (asymmetric) variant of hypsarrhythmia. During video EEG, both focal impaired awareness seizures and typical infantile spasms were recorded. Despite achieving therapeutic serum levels of valproate (VPA, 97 mcg/mL), the patient did not experience a reduction in seizure frequency. Vigabatrin (VGB) was introduced as an additional treatment, yet only a partial reduction in seizures was attained. Following the introduction of the third medication, lamotrigine (LTG), a complete cessation of seizures was observed for several weeks. At the age of 11th months, the patient was readmitted to the neurology unit, due to poor seizure control. A neurological examination revealed microcephaly (head circumference of 43 cm, below 3rd percentile according to [8]). Neither visual contact nor a social smile had been achieved yet. Fine and gross motor skills showed no progress; the patient was unable to control his head or roll from prone to supine position. Video EEG showed a marked improvement in comparison to the previous examination. MRI revealed extensive hyperintensities in both T2-weighted and FLAIR images of the thalami, globi pallidi and dorsal pons. The LTG dose was increased, resulting in a reduction in seizures. Thus, a gradual reduction in the VPA dose was ordered. At the point the present report was finalized (at 18 months), the patient was under the care of an outpatient neurological and palliative care team and was undergoing daily physiotherapy. A swallowing therapy programme was initiated under the supervision of a speech and language therapist. The patient was still fed via a nasogastric tube, but PEG tube implementation was considered. The patient remained on a course of treatment comprising human recombinant growth hormone and antiepileptic drugs.

## 3. Genetics Study Results

The patient was referred for a genetic consultation. The primary diagnosis was Prader–Willi syndrome. DNA methylation analysis by methylation-specific MLPA was performed as first-tier testing (SALSA^®^ MLPA^®^ Probemix ME028 Prader–Willi/Angelman, MRC Holland, Amsterdam, The Netherlands), as this detects copy number variations and the methylation status of the 15q11 region). A microdeletion of the analyzed region was excluded, while an abnormal DNA methylation pattern in the critical PWS region 15q11-q13 was identified. The analysis of microsatellite polymorphism in the 15q region in the proband and in both parents confirmed the maternal uniparental disomy of chromosome 15. Furthermore, SNV analysis indicated the presence of a mixed-type disomy, characterized by isodisomy of the 15q21 region and heterodisomy of the remaining part of chromosome 15.

The clinical presentation could not be fully explained by the PWS diagnosis alone. Consequently, genetic testing was extended to whole-exome sequencing (WES) as a second-tier approach using the NGS method. Trio-WES and an analysis of the mitochondrial genome was performed on the proband and his parents using the Twist Human Core Exome 2.0 and Twist mtDNA Panel (Twist Bioscience, San Francisco, CA, USA), in accordance with the manufacturer’s instructions (read depth 98.2—ge20; 98.4—ge10; median 117.0). The patient was found to be homozygous for a variant in the *GNB5* gene (hg38, 15:052133444-C>T, NM_016194.4:c.797G>A, p.(Trp266Ter)). The c.797G>A variant results in a stop gained change involving the alteration of a conserved nucleotide. The variant allele was identified at a frequency of 0.0000012 in 831,780 control chromosomes within the GnomAD database. The variant was located in the region of isodisomy on chromosome 15. The WES analysis of the maternal sample established her status as a carrier. The variant c.797G>A in the *GNB5* gene has not been previously reported in the literature. The c.797G>A variant is predicted to be pathogenic (10 points) according to the VarSome American College of Medical Genetics Classification [9].

## 4. Discussion

### 4.1. Prader–Willi Syndrome

Prader–Willi syndrome (PWS) is a rare neurodevelopmental contiguous gene syndrome caused by the loss of function of paternal copies of maternally imprinted genes located in the q11-q13 region of chromosome 15 [1,2,10]. This relatively well-described syndrome is a prototypical genetic disorder related to the imprinting mechanism [1,10]. Typically, patients with PWS present with severe hypotonia and feeding difficulties in early infancy, followed by hyperphagia resulting in obesity in early childhood, if not controlled [1]. Global developmental delay with behavioral disturbances, intellectual disability, hyperinsulinemia, growth hormone deficiency, hypothalamic hypogonadism, cryptorchidism and a short stature are common [6,11,12]. The etiology of PWS is complex and includes the following factors: paternal 15q11-q13 deletion, maternal chromosome 15 uniparental disomy (UPD15), Prader–Willi Syndrome/Angelman Syndrome (PWS/AS) imprinting defects, as well as complex chromosomal rearrangements [1,7,13]. A maternal UPD15-related etiology, the second most common causative mechanism of PWS, carries the risk of comorbid autosomal recessive (AR) disorders if the mother carries pathogenic variants in one of the genes on chromosome 15 associated with autosomal recessive traits [3,10,13].

The co-occurrence of PWS and additional autosomal recessive conditions (e.g., congenital ichthyosis linked to a homozygous pathogenic variant in the ceramide synthase 3 (*CERS3*) gene and hereditary spastic paraplegia type 11 linked to the spatacsin (*SPG11*) gene, STRC/CATSPER2-deletion-mediated deafness/infertility syndrome, Bloom syndrome and Tay–Sachs disease) has been previously reported in the literature [14,15]. It is important for physicians to consider the possibility of such a scenario when encountering an unusual or unexpectedly evolving phenotype in a PWS patient [3]. Prader–Willi syndrome is not considered as a frequent risk factor for epilepsy, unlike its related imprinting disorder, Angelman syndrome [16,17]. The majority of PWS patients with epilepsy show a favorable evolution and their seizures tend to be easy to control with monotherapy [18,19]. EEG patterns lack specificity, though hypsarrhythmia or burst–suppression patterns are uncommon [18,20,21]. The presence of severe epileptic encephalopathy with hypsarrhythmia and infantile spasms is not consistent with the typical PWS phenotype [18,20]. A conventional structural MRI of a PWS patient typically reveals general cortical atrophy and gyrification, corpus callosum agenesis, cerebellar abnormalities, light ventriculomegaly and pituitary hypoplasia, as well as Sylvian fissure abnormalities with incomplete insula closure [20]. Corpus callosum (CC) agenesis is a common occurrence in cases of hereditary intellectual impairment. Cerebellar abnormalities may be associated with hypotonia, while Sylvian fissure abnormalities may be linked with language impairment [21]. Pituitary hypoplasia may be associated with endocrine sequelae of PWS [21,22]. However, MR imaging results are not specific for PWS and vary among studies [23].

### 4.2. GNB5-Related Neurodevelopmental Disorder

The *GNB5* gene encodes the guanine nucleotide-binding protein, which is involved in inhibitory G protein signalling. G protein-coupled signalling plays a pivotal role in neuronal communication, including the regulation of the autonomous nervous system [24,25,26].

In their original report published in 2016, Lodder and Merla identified nine affected individuals (six females and three males, aged 6 to 23 years) from six unrelated families presenting with cognitive deficits, delayed motor development and cardiac conduction defects. All subjects were found to have variants in *GNB5* and presented with a similar rare phenotype, which was later named *GNB5*-related neurodevelopmental disorder [27,28,29].

Lodder–Merla syndrome types 1 and 2 are two eponymous phenotypes within the *GNB5*-related disorder spectrum. Lodder–Merla syndrome type 1 (LDMLS1) is characterized by delayed psychomotor development, profound intellectual disability, absent speech, bradycardia and other cardiac arrhythmias. The phenotypic spectrum also encompasses visual abnormalities, epilepsy, hypotonia, and gastric reflux. LDMS1 is recognized in individuals homozygous for null alleles in the *GNB5* gene, as demonstrated by Poke and Sciacca [29,30]. In contrast, Lodder–Merla syndrome type 2 (LDMLS2) is a less severe phenotype, characterized by sinus node dysfunction in combination with variable intellectual and language disabilities, attention deficit hyperactivity disorder, and impaired fine motor skills. In some LDMLS2 patients, cognitive development is normal [28,29,30].

A different proposed nosology classifies the severe phenotype of *GNB5*-related neurodevelopmental disorder as an intellectual developmental disorder with cardiac arrhythmia (IDDCA, OMIM 617173, ORPHA: 542306), which is consistent with LDMLS1; meanwhile, the milder disorder, which manifests as language delay, attention deficit/hyperactivity disorder, and cognitive impairment with or without cardiac arrhythmia (LADCI; OMIM: 617182), is consistent with LDMLS2 [29,30,31,32]

### 4.3. Neurodevelopmental Issues and Epilepsy in Patients with GNB5-Related Neurodevelopmental Disorder

The presentation of developmental delay in the first months of life in patients with *GNB5*-related neurodevelopmental disorder is associated with an unfavorable prognosis, characterized by severe or profound ID and, in some cases, epilepsy [29]. The five children with mild-to-moderate ID reported by Lodder, Malerba and DeNittis [27,31,33] were all nonverbal or used only a few spoken words, and had impaired fine motor skills. Hypotonia and hyporeflexia are commonly reported in severely affected patients. Eventually, hypertonia, spasticity and joint contractures may develop in a minority of cases [34].

Four of the nine patients from the original Lodder and Merla report had epilepsy [27,29]. To date, 26 of the 41 reported individuals have been diagnosed with developmental and epileptic encephalopathy, with marked developmental delay prior to the onset of epilepsy [27,29,35] The median age at seizure onset was six months (range: one week to three years). The semiology of the seizures was variable, comprising infantile spasms, focal motor seizures and tonic–clonic seizures. The most prevalent seizure type appeared to be infantile spasms [35]. EEG patterns could be normal in the early infancy; however, a burst suppression pattern (at 2 to 5 months) or hypsarrhythmia (at 2 months to 3 years) developed later. By the age of three, the EEG pattern had evolved to multifocal discharges with background slowing [29,35]. To date, no data on pathognomonic neuroimaging abnormalities in *GNB5*-related disease is available [29,31,34].

### 4.4. Cardiac Issues

The *GNB5* gene plays a key role in the parasympathetic regulation of heart rate [35,36]. Cardiac arrhythmia with bradycardia and ectopic beats represents a core symptom in *GNB5* knock-out mouse models. Sick sinus syndrome and the resulting bradycardia, usually with a preserved positive chronotropic response to stress, is the most common arrhythmia in humans [29,30,31,32,36]. The patient may be asymptomatic or may present with cyanosis or apnea during long periods of asystole. It is of the utmost importance not to misinterpret paroxysmal cyanotic and apneic episodes due to cardiac rhythm abnormalities as an epileptic ictal presentation [37,38]. The implementation of proarrhythmic antiepileptic drugs (e.g., those that prolong the QT interval or induce bradycardia) in this situation can have life-threatening consequences for the patient [39,40]. It is noteworthy that the presence or absence of cardiac involvement does not correlate with the severity of developmental impairment [29]. Other arrhythmias, including atrioventricular block, atrial tachycardia, atrial fibrillation and sinus tachycardia, have been observed in individual cases [29,41]. In the case reports by Lodder, Vernon and Yazdani, six children had a pacemaker implanted, with two of them having symptomatic bradycardia [27,42,43].

### 4.5. Eye Issues

Visual impairment is common in children with the severe phenotype [29,31,34,42]. Strabismus, myopic refractive errors and horizontal or vertical nystagmus primarily affect children with the severe phenotype. Fundus changes are rare, with optic atrophy or disc pallor being reported in individual cases. Retinopathy is associated with phototransduction recovery disarrangement in both rod and cone photoreceptors (bradyopsia) and rod ON-bipolar cell dysfunction [44,45]. At present, little is known about the genotype–phenotype correlation in visual impairment [29,45,46].

## 5. Conclusions

Uniparental disomy is associated with two principal risks: the inheritance of a recessive trait and the occurrence of an imprinting disorder [1,3]. In some cases, these two risks may affect a single patient, resulting in a rare comorbidity. It is generally accepted that comorbidities are associated with poorer health outcomes and more complex clinical management [47]. In the case of rare diseases, they may further delay proper diagnosis and treatment, as the clinical picture is blurred [3]. This is corroborated by the case of the boy presented in this paper, who was eventually diagnosed with two rare diseases: Prader–Willi syndrome due to maternal UPD15 and autosomal recessive Lodder–Merla syndrome caused by a novel pathogenic variant in the *GNB5* gene. There are three main lessons to be learnt from his case. First, the occurrence of unexpected or severe phenotype features in PWS patients should always prompt the consideration of a comorbid genetic disease attributed to genes located in the Prader–Willi syndrome-associated critical region of chromosome 15q [1,5,20] or those located elsewhere on chromosome 15, where a pathogenic variation would normally follow recessive inheritance. From a clinical perspective, in our case, epileptic encephalopathy, early profound neurodevelopmental disability and visual system involvement could not be fully explained by the Prader–Willi diagnosis alone. Second, in the case of epileptic encephalopathy with cardiac arrythmia, cardiologist consultation and further genetic testing are necessary in order to diminish the risks associated with untreated arrhythmia and ensure proper and safe anti-epileptic therapy [35,37,47]. Third, the case provides further support for the hypothesis that uniparental disomy may cause Lodder–Merla syndrome. This highlights the importance of parental testing and comprehensive genetic testing to elucidate the underlying genetic etiology of the disease, thereby facilitating an accurate assessment of the recurrence risk.

## Data Availability

The original contributions presented in this study are included in the article. Further inquiries can be directed to the corresponding authors.

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
