# Peer review of "Mixed Segmental Uniparental Disomy of Chromosome 15q11-q1 Coexists with Homozygous Variant in GNB5 Gene in Child with Prader–Willi and Lodder–Merla Syndrome"

_genes, 2025, doi:10.3390/genes16060689_

Round 1
Reviewer 1 Report
Comments and Suggestions for Authors
PWS and Lodder-Merla synd; 15q11 UPD genes-3663158;
May 2025 Reviewer Comments
This is a well-constructed and interesting case report of a single patient with Lodder-Merla syndrome arising from isodisomy of the GNB5 gene located at 15q21.2, recognised as a comorbidity to their concomitant presenting diagnosis of Prader-Willi syndrome. It highlights the need to consider the possibility of a concomitant autosomal recessive condition mapping to the same chromosome, in any patient with an imprinting disorder arising from complete or partial isodisomy of a chromosome.
Major Point
However this generalisation aspect does need to be developed further in this paper. In particular the authors refer in their conclusion (page 6, lines 243-253) to the three main lessons of this case as:
i) An unexpected phenotype in PWS should prompt consideration of a comorbid genetic disease attributed to genes located in the Prader-Willi syndrome-associated critical region of chromosome 15q [6, 8, 20].
ii) Need for cardiology assessment and genetic testing in cases of epileptic encephalopathy with cardiac arrythmia
iii) UPD can be a cause of Lodder-Merla syndrome
However, if the GNB5 gene maps to 15q21.2, and the Prader Willi syndrome critical region is 15q11-13, the first of these conclusions would not seem to be strictly accurate. Rather the case example illustrates that a comorbid condition (which would usually follow recessive inheritance) can be present in a patient with PWS caused by UPD where the gene underlying that condition maps anywhere on chromosome 15. The ‘first lesson’ therefore needs to be amended accordingly.
It is a separate 4th potential mechanism (but not strictly a lesson from this case) that genes within the PWS-critical region can show imprinting dysfunction (and consequent additions to the phenotype) in cases of PWS.
Accordingly, while the authors have mentioned (page 4, line 141-4) two other genetic conditions that can present as comorbidities in PWS arising from UPD (ie. CERS3-associated congenital ichthyosis type9 , and HSP11), this case report could serve as an opportunity to list all the different normally-recessively-inherited conditions whose genes are located on chromosome 15 which have been reported to have been observed in patients with PWS (perhaps and/or Angelman syndrome) due to isodisomy. This would reinforce the main lesson to be drawn from this paper, and therefore significantly increase its main cautionary message and therefore value to paediatric practice in general. Eg. see Woodage T, et al Am J Hum Genet. 1994 Jul;55(1):74-80.
There are also several minor points, many of which are minor corrections to the English.
These are listed in the Table below by Page and Line number. The major point above is also mentioned as points 18 and 26.
Table of more minor points
No. |
Page; Line |
Current text |
Suggested revision |
Comment |
1 |
Abstr P1,L25 |
linked to novel |
linked to a novel |
|
2 |
P1,L29 |
consultation with a cardiologist consultation |
consultation with a cardiologist |
Delete the 2nd ‘consultation’ |
3 |
P1,L34-5 |
familial cascade testing to establish the recurrence risk [2, 3]. |
This statement may need reconsideration, depending on local or national guidelines. |
In rare recessively- inherited conditions, with family-specific pathogenic variants (ie. no one variant accounts for a sizeable proportion of cases), cascade testing only becomes relevant where there is consanguinity, or the likelihood of close common ancestral background between partners. Otherwise, it would become akin to population screening for recessive disorders, since we all carry at least one pathogenic recessive variant. |
4 |
Case Report P2,L53 |
unrelated, healthy parents was |
unrelated, healthy parents was |
Remove the italics |
5 |
P2,L54 |
at 38th week of gestation |
At 38 weeks gestation OR at the 38th week of gestation |
|
6 |
P2,L54 |
breach |
breech |
|
7 |
P2,L58-9 |
with an Apgar score of 7 points at the first minute and 10 after tenth mi- 58 nute of life. |
with an Apgar score of 7 at 1 minute and 10 at 10 minutes. OR with Apgar scores of 7 and 10. |
This sentence can legitimately be shortened. |
8 |
P2,L59 |
Life. Bilateral A physical |
Life. A physical |
Delete this ‘bilateral’ |
9 |
P2,L64-5 |
bilateral congenital pneumonia |
bilateral pneumonia |
Was the pneumonia ‘congenital’ – ie. present from birth, or acquired in the first few days of life ? |
10 |
P2,L68 |
infantile spasms. |
Infantile spasms (with hypsarrythmia on EEG)
OR: ‘Infantile seizures’ |
Infantile spasms are a specific type of seizure characterised by hypsarrythhmia – it would be helpful to mention the EEG here as well – otherwise at this point in the text they may be best termed ‘infantile seizures’ |
11 |
P2,L72 |
Subsequently…. …an anterior fontanelle measuring 2.5x3cm |
Subsequently… …., and an enlarged anterior fontanelle measuring 2.5x3cm at ** weeks. |
It is important to know the baby’s age when giving the fontanelle size (and to say that it is enlarged).
|
12 |
P2,L73 |
light, bilateral |
light, and bilateral |
|
13 |
P2,L79 |
registered atypical (asymmetric) variant |
registered an atypical (asymmetric) variant |
|
14 |
P2,L86 |
to neurology unit, |
to the neurology unit, |
|
15 |
P3, L96-7 |
Patient was still fed |
The patient was still fed |
|
16 |
P3,L103 |
The microdeletion |
A microdeletion |
|
17 |
P3,L114 |
homozygous homozygous |
homozygous |
The word: ‘homozygous’ has been duplicated in error. |
18 |
P4,L141-4 |
The co-occurrence of PWS and additional autosomal recessive conditions… ….has been previously reported in literature [14,15]. |
The co-occurrence of PWS and additional autosomal recessive conditions… …has been previously reported in the literature [14,15]. |
The authors may like to consider other examples as well, as it would be helpful to use this case report to provide a comprehensive list (as a Table) of all reported examples of ‘autosomal recessive’ disomy co-existing with PWS (or with Angelman synd.) For example: see Woodage T, et al Am J Hum Genet. 1994 Jul;55(1):74-80. |
19 |
P4,L159 |
sequela |
sequelae |
Should be plural here ? |
20 |
P4,L186 |
consistent with |
is consistent with |
|
21 |
P5,L206 |
By the age of three, EEG pattern had evolveds to… |
By the age of three, the EEG pattern had evolved to… |
|
22 |
P5, L210 |
has key role |
has a key role |
|
23 |
P5,L214 |
may presents with |
may present with |
|
24 |
P5,L220 |
that the, presence |
that the presence |
The comma can be removed |
25 |
P5,L223 |
reports reports |
reports |
|
26 |
P5,L245-7 |
of a comorbid genetic dis ease attributed to genes located in the Prader-Willi syndrome-associated critical region of chromosome 15q [6, 8, 20]. |
of a comorbid genetic dis ease attributed to genes located either in the Prader-Willi syndrome-associated critical region of chromosome 15q [6, 8, 20], or elsewhere on chromosome 15 where pathogenic variation would normally follow recessive inheritance. |
It is important to stress that the gene doesn’t have to be in the PWS critical region, but can be elsewhere on chr 15 where a normally recessively-inherited condition can manifest if it is in the regions that happen to be isodisomic in that particular patient when there is UPD. This is an important concept, and hence the recommendation in point 18 above, which becomes a point for major revision of this paper. |
Comments on the Quality of English Language
There are several places where the |English language can be improved, although most of these involve adding the articles 'a' or 'the' before some nouns. The most obvious examples are listed in the Table of minor corrections.
The accuracy of proof-checking this manuscript before submission is also a little disappointing, as there are a few places where a word has been duplicated in error. These are also listed in the minor corrections table.
Author Response
Thank you very much for taking the time to review our manuscript.
Please find the detailed responses in the attached .doc file.
At point number 11 we have changed sentence to: "Subsequently… …., an anterior fontanelle measuring 2.5x3cm at four months of age"
Kind regards,
Robert Åšmigiel, Tomasz Marczyk

Reviewer 2 Report
Comments and Suggestions for Authors
The manuscript examines the case of a 14-month-old male infant that had a dual diagnosis of Prader-Willi syndrome (PWS) and Lodder-Merla syndrome.
I have some comments that would strengthen the article.
Main comments
1, The abstract should not have references.
2, The names of genes should be spelled out before they are abbreviated i.e. line 48.
3, References should be consecutively numbered and appear in sequence. For example, in the introduction reference 6 and 7 (line 42) appear before reference 1 etc.
4, I would suggest that the authors amend the abstract to clearly emphasize that this is a case report.
5, The manuscript would benefit from English language editing i.e. line 25 ‘detiledin’, hyperlinks in line 162.
Minor comment
6, Gene names ought to be in italics i.e. line 115.
Title
7, The manuscript is a case study, and this should be reflected in the main title.
Abstract
8, Line 21-22: The authors should amend the following text “Managing comorbidities associated with rare diseases presents unique clinical challenges”, while this may be true, reference 3 is about uniparental isodisomy and does not account for rare diseases arising out of de novo mutations. I would suggest using a more appropriate reference.
9, It is unclear from the abstract whether the co-occurring diagnosis of PWS is with LDMS1 (severe phenotype) or LDMLS2 (milder phenotype).
Introduction
10, Line 40: “rare but relatively well known”? Please consider amending this.
11, Line 44 – rather than ‘paper’ suggest writing ‘case report’.
12, Line 48 – please spell out the gene and provide information about the mutation.
13, Line 51 – the authors should clarify whether the case report has been described elsewhere. I note that there is already an abstract freely available:
Beata et al. Mixed segmental uniparental disomy of chromosome 15q11-q1 coexists with homozygous variant in GNB5 gene in child with Prader-Willi and Lodder-Merla syndrome. Endocrine Abstracts. 10.1530/endoabs.110.EP752
Case Report
14, Line 79 – this type of EEG pattern (hypsarrhythmia) is seen in West Syndrome. Was this considered as a differential diagnosis?
Genetics Study Results
15, There should be some mention of the methods here i.e. was NGS performed?
16, Line 102 - How was methylation specific MLPA performed as first tier testing? Please specify MLPA as this may not be obvious to the lay reader. What did the methylation pattern show?
17, Line 117-118: This sentence should be explained better. In the context of the genetic findings what is meant by “no instances of homozygosity”?
Discussion
18, Line 125 - OMIM should be inserted into line 40 where PWS is first mentioned (line 40).
19, Section 4.1 has overlap with the introduction and some repetition. Please consider amending.
20, It is quite difficult to determine what symptoms were present in the actual case. For example, cardiac, eye and epilepsy are described as problems with GNB5 mutation but are these problems associated with the case presented?
Conclusions
21, What is the estimated occurrence of the dual diagnosis for Prader-Willi and Lodder-Merla Syndrome?
22, From a clinical perspective what would be the key symptoms that would also prompt an investigation for suspected Lodder-Merla Syndrome? I think this information would be useful to include.
Comments on the Quality of English LanguageThe manuscript would benefit from English language editing.
Author Response
Thank you very much for taking the time to review our manuscript.
Please find the detailed responses below.
Main comments
1, The abstract should not have references. [We have removed references from the abstract]
2, The names of genes should be spelled out before they are abbreviated i.e. Line 48. [We have spelled out name of genes e.g. GNB5, CERS3, SPG11 used in text]
3, References should be consecutively numbered and appear in sequence. For example, in the introduction reference 6 and 7 (line 42) appear before reference 1 etc. [We have corrected this, numbered references appear in text in sequence]
4, I would suggest that the authors amend the abstract to clearly emphasize that this is a case report. [We have changed sentence „The existence of such phenomena is evidenced by the case of a boy who was ultimately diagnosed with two rare diseases” page 1, lines 29-30 into „ The existence of such phenomena is evidenced by our case report of a boy who was ultimately diagnosed with two rare diseases”]
5, The manuscript would benefit from English language editing i.e. line 25 ‘detiledin’, hyperlinks in line 162. [We corrected the spelling error and removed hyperlink]
Minor comment
6, Gene names ought to be in italics i.e. Line 115. [We have written all gene names in italics]
Title
7, The manuscript is a case study, and this should be reflected in the main title. [Character of this research was mentioned in line 1 on page 1 above the title and in the abstract and on page two in line 56 in introduction]
Abstract
8, Line 21-22: The authors should amend the following text “Managing comorbidities associated with rare diseases presents unique clinical challenges”, while this may be true, reference 3 is about uniparental isodisomy and does not account for rare diseases arising out of de novo mutations. I would suggest using a more appropriate reference. [We have removed references from the abstract as mentioned above in point 1 and changed numeration of references]
9, It is unclear from the abstract whether the co-occurring diagnosis of PWS is with LDMS1 (severe phenotype) or LDMLS2 (milder phenotype). [We have changed sentence in line 29-33 on page 1 „The existence of such phenomena is evidenced by our case report of a boy who was ultimately diagnosed with two rare diseases: Prader-Willi syndrome...., and autosomal recessive Lodder-Merla syndrome....” to „The existence of such phenomena is evidenced by our case report of a boy who was ultimately diagnosed with two rare diseases: Prader-Willi syndrome... and autosomal recessive Lodder-Merla type 1 syndrome...”
Introduction
10, Line 40: “rare but relatively well known”? [We have changed this sentence fragment to „well-known neurodevelopmental...”]
11, Line 44 – rather than ‘paper’ suggest writing ‘case report’. [We have changed „paper” to „case report”]
12, Line 48 – please spell out the gene and provide information about the mutation. [We spelled out gene name as requested. We provided information about mutation in line 150-151, page 4, section „molecular analysis”].
13, Line 51 – the authors should clarify whether the case report has been described elsewhere. I note that there is already an abstract freely available:
Beata Wikiera et al. Mixed segmental uniparental disomy of chromosome 15q11-q1 coexists with homozygous variant in GNB5 gene in child with Prader-Willi and Lodder-Merla syndrome. Endocrine Abstracts. 10.1530/endoabs.110.EP752
[This abstract refers to the same patient and was described by the same authors. Abstract mentioned above was prepared for poster session of 27th Symposium of Polish Association of Pediatric Endocrinology and Diabetology]
Case Report
14, Line 79 – this type of EEG pattern (hypsarrhythmia) is seen in West Syndrome. Was this considered as a differential diagnosis? [EEG registered an atypical (asymmetric) variant of hypsarrhythmia. During video EEG both focal impaired awareness seizures and typical infantile spasms were recorded. We considered West syndrome in differential diagnosis. In fact, boy appeared to fulfill the classic criteria of West syndrome i.e. infantile spasms, hypsarrythmia and neurodevelopmental disability. However severe clinical presentation, partial complex seizurez and profound disabilitiy before appearance of infantile spasms opposed hypotesis of idiopathic West syndrome and prompted further investigation. We haven't included special section about differential diagnosis in our paper]
Genetics Study Results
15, There should be some mention of the methods here i.e. was NGS performed?
[we add to the text: „The clinical presentation could not be fully explained by the PWS diagnosis alone. Consequently, genetic testing was extended to whole-exome sequencing (WES) as a second-tier approach using NGS method.”]
16, Line 102 - How was methylation specific MLPA performed as first tier testing? Please specify MLPA as this may not be obvious to the lay reader. What did the methylation pattern show?
[we add to the text: „The primary diagnosis was Prader-Willi syndrome. DNA methylation analysis by methylation-specific MLPA was performed as a first-tier testing (SALSA® MLPA® Probemix ME028 Prader-Willi/Angelman which detects copy number variations and methylation status of the 15q11 region)”]
17, Line 117-118: This sentence should be explained better. In the context of the genetic findings what is meant by "no instances of homozygosity"?
["no instances of homozygosity" means that there is no description in literature of homozygosity for mentioned variant, but we remove this part of sentenece]
Discussion
18, Line 125 - OMIM should be inserted into line 40 where PWS is first mentioned (line 40). [We have inserted OMIM numbers in lines where PWS and Lodder-Merla syndromes are first mentioned i.e. Page 1 line 51 and page 1 line 71, respectively]
19, Section 4.1 has overlap with the introduction and some repetition. [We have changed introduction to avoid overlapping with section 4.1].
20, It is quite difficult to determine what symptoms were present in the actual case. For example, cardiac, eye and epilepsy are described as problems with GNB5 mutation but are these problems associated with the case presented? [Our patient presented with cognitive deficits, delayed motor development and epilepsy. Visual system involvement was represented by no visual contact, no fixation, nystagmus, slow pupils reaction to light; boy had no bradycardia or other cardiac arrythmias]
Conclusions
21, What is the estimated occurrence of the dual diagnosis for Prader-Willi and Lodder-Merla Syndrome?
[Our case is the first case described in literature. We mentioned about it in Introduction.]
22, From a clinical perspective what would be the key symptoms that would also prompt an investigation for suspected Lodder-Merla Syndrome? I think this information would be useful to include. [We have added sentence: „From the clinical perspective in our case epileptic encephalopathy, early profound neurodevelopmental disability and visual system involvement could not be fully explained by the PWS diagnosis alone” in line 303, page 7, section conclusion; we also emphasized in next sentence that „... in case of epileptic encephalopathy with cardiac arrythmia, cardiologist consultation and further genetic testing are necessary”].
Thank you again for taking the time to review our manuscript.
We are waiting further suggestions.
Kind regards,
Robert Åšmigiel, Tomasz Marczyk

Reviewer 3 Report
Comments and Suggestions for Authors
Review of Mixed Segmental Uniparental Disomy of Chromosome 15q11-q1 Coexists with Homozygous Variant in GNB5 Gene in Child with Prader-Willi and Lodder-Merla Syndrome, by Marczyk Tomasz and colleagues (Genes-3663158)
This manuscript describes a patient with a combination of Prader-Willi syndrome due to maternal UPD15, and Lodder-Merla syndrome linked and a novel pathogenic variant in the GNB5 gene. Since such a type of a double diagnosis has not yet been reported, this case report in highly worthwhile. A few items need attention, however.
In line 57 the weight of the patient at birth is given as 2110 grams. To judge this value it would be necessary to know what percentile of the local newborns this represents. Other observations at birth (e.g. Apgar scores, birth height, OFC) are also welcome.
The WES procedure needs few more details. What was the minimal (and maybe also median or average) read depth? Also details regarding the discovery and interpretation of variants (number of VUS, tools used for variant triage) should be given.
A figure with an IGV or similar plot of the WES data for the GNB5 gene would improve this report.
In line 113 the word homozygous was repeated. Please check the manuscript for possible other typing errors.
Author Response
Thank you very much for taking the time to review this manuscript. Please find the our responses below.
Comment: In line 57 the weight of the patient at birth is given as 2110 grams. To judge this value it would be necessary to know what percentile of the local newborns this represents. Other observations at birth (e.g. Apgar scores, birth height, OFC) are also welcome.
Answer: We have added mentioned information. page 2, lines 85-89 "The infant weighed 2110 grams at birth (below 3rd percentile according to WHO growth standards [8]), with length 48cm (15th percentile according to [8]), head circumference 34 cm (15-50 percentile according to [8], with an Apgar score of 7 at 1 minute and 10 at 10 minutes."
Comment:
The WES procedure needs few more details. What was the minimal (and maybe also median or average) read depth? Also details regarding the discovery and interpretation of variants (number of VUS, tools used for variant triage) should be given. A figure with an IGV or similar plot of the WES data for the GNB5 gene would improve this report.
Answer:
We add new information to the text (page 4, lines 151-155) „Trio-WES and an analysis of the mitochondrial genome was performed on the proband and his parents using the Twist Human Core Exome 2.0 and Twist mtDNA Panel (Twist Bioscience, San Francisco, CA, USA), in accordance with the manufacturer’s instructions (read depth 98,2 – ge20; 98,4 – ge10; median 117.0).” In our opinion in clinical manuscript is no space for molecular figure. Please accept this answer.
Kind regards,
Robert Åšmigiel, Tomasz Marczyk
Round 2
Reviewer 1 Report
Comments and Suggestions for Authors
PWS and Lodder-Merla synd; 15q11 UPD genes-3663158; Revision 1
29.05.25 Reviewer Comments
In this revised version of this paper, the authors have attended to the comments made in the review of the original, although should still try to give a comprehensive list of other recessively inherited conditions which have been reported with PWS arising from UPD of chromosome 15, as indicated in Point 9 in the Table below. However, this , and the other outstanding corrections needed can all be considered as ‘minor edits’
The paper will be a useful addition to the PWS and UPD literature
Table of outstanding minor corrections
No. |
Page; Line |
Current text |
Suggested revision |
Comment |
1 |
P1,L19 |
to condition |
to the condition |
|
2 |
P1, L30 |
Unusual or severe phenotype |
An unusual or severe phenotype OR Unusual or severe phenotypes |
|
3 |
P1, L32 |
located in PWS critical region |
located in the PWS critical region, or elsewhere on chromosome 15. |
But also, it isn’t just genes located in the critical region that need consideration; but any gene on chromosome 15 in which pathogenic variants can underlie conditions following autosomal recessive inheritance. This sentence must be expanded accordingly. |
4 |
P2, 46 |
is well-known |
is a well-known |
|
5 |
P2, L52 |
Boy |
The boy |
|
6 |
P2, L73 |
which was subsequently diagnosed |
And the baby was subsequently diagnosed |
Reduced fetal movement wouldn’t subsequently be diagnosed as IUGR. |
7 |
P2, L91 |
limb segments, an anterior fontanelle |
limb segments, and an anterior fontanelle |
|
8 |
P3, L143 – P4,L144 |
Reported in literature |
Reported in the literature. |
|
9 |
P4, L164-8 |
The co-occurrence of PWS and additional autosomal recessive conditions (e.g. congenital ichthyosis linked to homozygous pathogenic variant in the seramide synthase 3 (CERS3) gene and hereditary spastic paraplegia type 11 linked to the spatacsin (SPG11) gene and Bloom syndrome) has been been previously reported in the literature [14, 15, 16]. |
The co-occurrence of PWS arising from UPD, and additional autosomal recessive conditions on chromosome 15 (e.g. congenital ichthyosis linked to a homozygous pathogenic variant in the seramide synthase 3 (CERS3) gene, hereditary spastic paraplegia type 11 linked to the spatacsin (SPG11) gene, STRC/CATSPER2-deletion-mediated deafness/infertility syndrome, Bloom syndrome, and Tay-Sachs Disease) have all been previously reported in the literature [14, 15, 16]. |
The authors could have dealt more fully with the issue of PWS-associated recessive conditions arising from UPD, as raised in this reviewer’s first report. If the authors are not producing an updated Table similar to the Table in Reference 14 (Muthusamy et al), they should at least list all 3 of the additional conditions mentioned in that paper, in order to help improve the consideration and hence recognition of these in clinical practice which involves PWS patients. |
10 |
P5,L207 |
…ORPHA:542306) is consistent with LDMLS1, |
…ORPHA:542306), which is consistent with LDMLS1, |
|
11 |
P5,L209 |
MIM: 617182) consistent with LDMLS2 [29, 30, 31, 32] |
MIM: 617182) is consistent with LDMLS2 [29, 30, 31, 32] |
|
12 |
P6, L278 |
accurateassessment |
accurate assessment |
|
Comments on the Quality of English Language
There are still a few places where the English language can be improved, although mostly involve adding the articles 'a' or 'the' before some nouns, or editing the punctuation. I have tried to alert the authors to the most obvious of these in the Table of minor comments.
Author Response
Dear Reviewer,
Thank you very much for taking the time to review our manuscript again.
First, we enumerated recessively inherited conditions which have been reported with PWS arising from UPD15, according to your comment and paper by Muthusamy (i.e. congenital ichthyosis linked CERS3 gene, hereditary spastic paraplegia type 11, STRC/CATSPER2-deletion-mediated deafness/infertility syndrome, Bloom syndrome and Tay-Sachs disease).
Second, we corrected minor errors as shown in the table below.
No. |
Page; Line |
Current text |
Suggested revision |
|
1 |
P1,L19 |
to condition |
to the condition |
We corrected this error. |
2 |
P1, L30 |
Unusual or severe phenotype |
An unusual or severe phenotype OR Unusual or severe phenotypes |
We corrected this error. |
3 |
P1, L32 |
located in PWS critical region |
located in the PWS critical region, or elsewhere on chromosome 15. |
We corrected this error and expanded the sentence. |
4 |
P2, 46 |
is well-known |
is a well-known |
We corrected this error. |
5 |
P2, L52 |
Boy |
The boy |
We corrected this error. |
6 |
P2, L73 |
which was subsequently diagnosed |
And the baby was subsequently diagnosed |
We corrected this error. We change this sentence to: "The mother observed reduced fetal movement. The baby was subsequently diagnosed with intrauterine growth restriction based on ultrasound-estimated fetal measurements." |
7 |
P2, L91 |
limb segments, an anterior fontanelle |
limb segments, and an anterior fontanelle |
We corrected this error. |
8 |
P3, L143 – P4,L144 |
Reported in literature |
Reported in the literature. |
We corrected this error. |
9 |
P4, L164-8 |
The co-occurrence of PWS and additional autosomal recessive conditions (e.g. congenital ichthyosis linked to homozygous pathogenic variant in the seramide synthase 3 (CERS3) gene and hereditary spastic paraplegia type 11 linked to the spatacsin (SPG11) gene and Bloom syndrome) has been been previously reported in the literature [14, 15, 16]. |
The co-occurrence of PWS arising from UPD, and additional autosomal recessive conditions on chromosome 15 (e.g. congenital ichthyosis linked to a homozygous pathogenic variant in the seramide synthase 3 (CERS3) gene, hereditary spastic paraplegia type 11 linked to the spatacsin (SPG11) gene, STRC/CATSPER2-deletion-mediated deafness/infertility syndrome, Bloom syndrome, and Tay-Sachs Disease) have all been previously reported in the literature [14, 15, 16]. |
Thank you for that comment, we enumerated other recessively inherited conditions which have been reported with PWS arising from UPD15, according to your comment and paper by Muthusamy et al. |
10 |
P5,L207 |
…ORPHA:542306) is consistent with LDMLS1, |
…ORPHA:542306), which is consistent with LDMLS1, |
We corrected this error. |
11 |
P5,L209 |
MIM: 617182) consistent with LDMLS2 [29, 30, 31, 32] |
MIM: 617182) is consistent with LDMLS2 [29, 30, 31, 32] |
We corrected this error. |
12 |
P6, L278 |
accurateassessment |
accurate assessment |
We corrected this error. |
Kind regards,
Tomasz Marczyk, Robert Åšmigiel
Reviewer 2 Report
Comments and Suggestions for Authors
Thank you for making the changes. However, I do think that it should be stated in the text that the results of this case study have been described elsewhere albeit in a poster.
Comments on the Quality of English LanguageThe manuscript would benefit from English language editing.
Author Response
Thank you for you comment.
Accordingly, we added sentence “Case of this patient was presented during session of Joint Congress of the European Society for Paediatric Endocrinology (ESPE) and the European Society of Endocrinology (ESE) 2025 as a poster [7]“ in introduction (page 2, line 71-73). We also made revision of the spelling and grammar errors.
Kind regards
Robert Åšmigiel, Tomasz Marczyk